# Antibiotic resistant bacteria and commensal fungi are common and conserved in the mosquito microbiome

Josephine Hyde[1], Courtney Gorham[1], Doug E. Brackney[1,2], Blaire Steven[1]*

**1** Department of Environmental Sciences, Connecticut Agricultural Experiment Station, New Haven, Connecticut, United States of America, **2** Center for Vector Biology and Zoonotic Diseases, Connecticut Agricultural Experiment Station, New Haven, Connecticut, United States of America

* blaire.steven@ct.gov

**Data Availability Statement:** Representative sequences for each OTU (calculated used abundance weighted scoring) were uploaded to the NCBI Nucleotide database with accession numbers

## Abstract

The emerging and increasing prevalence of bacterial antibiotic resistance is a significant public health challenge. To begin to tackle this problem, it will be critical to not only understand the origins of this resistance but also document environmental reservoirs of antibiotic resistance. In this study we investigated the possibility that both colony and field caught mosquitoes could harbor antibiotic resistant bacteria. Specifically, we characterized the antibiotic resistant bacterial populations from colony-reared *Aedes aegypti* larvae and adults and two field caught mosquito species *Coquillettidia perturbans* and *Ochlerotatus canadensis*. The cultured bacterial populations were dominated by isolates belonging to the class Gammaproteobacteria. Among the antibiotic resistant populations, we found bacteria resistant to carbenicillin, kanamycin, and tetracycline, including bacteria resistant to a cocktail of all three antibiotics in combination. The antibiotic resistant bacteria were numerically rare, at most 5% of total cell counts. Isolates were characterized by 16S rRNA gene sequencing, and clustering into Operational Taxonomic Units (OTUs; 99% sequence identity). 27 antibiotic resistant OTUs were identified, although members of an OTU did not always share the same resistance profile. This suggests the clustering was either not sensitive enough to distinguish different bacteria taxa or different antibiotic resistant sub-populations exist within an OTU. Finally, the antibiotic selection opened up a niche to culture mosquito-associated fungi, and 10 fungal OTUs (28S rRNA gene sequencing) were identified. Two fungal OTUs both classified to the class Microbotryomycetes were commonly identified in the field-caught mosquitoes. Thus, in this study we demonstrate that antibiotic resistant bacteria and certain fungi are common and conserved mosquito microbiome members. These observations highlight the potential of invertebrates to serve as vehicles for the spread of antibiotic resistance throughout the environment.

## Introduction

Mosquitoes, like many complex organisms, live in association with a microbiome made up of bacteria, fungi and viruses [e.g. 1–5]. Descriptive studies of the mosquito microbiome suggest

MN046275- MN046301 for bacterial 16S rRNA genes and MN046918-MN046927 for fungal 28S rRNA genes.

**Funding:** This work was funded by the Louis A. Magnarelli Postdoc program administered by the CAES Board. The funder had no role in study design, data collection and analysis, decision to publish, or preparation of the manuscript.

**Competing interests:** The authors have declared that no competing interests exist.

that the bacterial species richness of the microbiome is relatively low, generally made up of 10 to 50 species [1,6]. As mosquitoes are vectors of blood borne arboviruses, such as West Nile virus, dengue virus, and chikungunya virus, the microbiome has received substantial research as a possible factor that could interact with mosquito vector competence [7,8]. Yet, these studies are hampered by an inability to systematically manipulate the microbiome.

The generation of axenic mosquitoes, allowing for establishing gnotobiotic hosts with a controlled microbiome, has only been reported recently [9,10]. Previous studies had relied on antibiotic treatment of the mosquitoes to reduce or "remove" the microbiome prior to experiments [11–14]. However, studies have documented that the mosquito microbiome can harbor antibiotic resistant populations [15,16] and thus antibiotic treatment is more likely to induce a dysbiosis in the microbiome rather than an elimination of the mosquitoes' microbiota [12]. Furthermore, antibiotic treatment can affect the host's physiology by inducing mitochondrial dysfunction and oxidative damage [17,18]. In this regard, it is often difficult to distinguish the effects of reducing the microbiome load via antibiotics versus potential effects of the antibiotics themselves.

In this study we set out to test how common antibiotic resistant bacteria are among colony-reared and field-caught mosquitoes. We hypothesized that the mosquito microbiome would naturally harbor antibiotic resistant bacterial populations. We further predicted that the wild-caught mosquitoes would harbor larger and more diverse antibiotic resistant populations due to their increased probability of encountering antibiotics in their environment, as the colony mosquitoes had no history of antibiotic treatment.

## Materials and methods

### Rearing and collection of colony mosquitoes

Colony reared mosquitoes (*Aedes aegypti*; Orlando strain) were maintained using standard conditions [9]. Larvae from the same egg hatch were collected from rearing pans with a Pasteur pipette and serially rinsed through three Petri dishes containing sterile water to remove external adhering bacteria. Adult female mosquitoes were collected 24 hours post blood feeding (hpbf) from females with a still visible blood bolus to ensure that the mosquito had blood fed, which insured the mosquitoes were in a similar physiological state.

### Testing colony-reared mosquitoes for antibiotic resistance genes

To investigate the presence of antibiotic resistance genes in the microbiome of colony-reared mosquitoes we employed a quantitative PCR array. Total DNA was extracted from a pool of three larvae and from the dissected midguts of three female adults 24 hpbf. DNA was extracted with the DNeasy PowerSoil kit (Qiagen) using standard protocols as previously described for mosquitoes [9]. The DNA was employed as a template for a Qiagen 96-well Microbial DNA qPCR Array (Antibiotic Resistant genes). This array detects 84 antibiotic resistant genes across multiple antibiotic classes and also includes two pan bacteria and positive PCR controls to test for the presence of inhibitors and PCR efficiency. A full list of the genes detected in the assay can be found on the manufacture's website. Quantitative PCR was performed on a BioRad C1000 Touch Thermocycler with CFX96 Real-Time System. Thermalcycling conditions consisted of 10 minutes at 95°C, followed by 40 cycles of: 15 seconds at 95°C and 2 minutes 60°C and a 1°C/s ramp rate (as recommended by the assay manufacture) with FAM fluorophore detection. Values for cycle quantification value ($C_q$) were recorded for each well and positive detections were based on the manufacture's recommendations.

## Field collection of mosquitoes

Mosquito trapping was conducted as described previously [19]. Briefly, $CO_2$ (dry ice) baited CDC miniature light traps equipped with aluminum domes were collected from six sites around Connecticut (Fig 1). Mosquitoes were collected as part of the state of Connecticut Mosquito Trapping and Arbovirus Surveillance Program. The Connecticut Mosquito Trapping and Arbovirus Surveillance Program was established by Public Act 97–289 in 1997. Traps were placed in the field in the afternoon, operated overnight, and retrieved the following morning. Adult mosquitoes were transported alive to the laboratory each morning in an ice chest lined with cool packs. Mosquitoes were immobilized with dry ice and transferred to chill tables where they were identified to species with the aid of a stereo microscope (90X) based on morphological characters. Individual females of two species, *Coquillettidia perturbans* and *Ochlerotatus canadensis* were identified and separated with sterile tweezers and processed for bacterial isolation immediately after collection from the field.

## Bacterial culturing and isolation

In initial experiments we tested the bacterial recovery from individual colony-reared and field-caught mosquitoes. For the colony mosquitoes an individual was sufficient to recover *c.a.* 1 x $10^7$ colony-forming units (CFUs; for larvae and post blood fed females) whereas individual field-caught mosquitoes displayed much lower CFUs, often with too few to reliably enumerate from an individual. For this reason, the field-caught mosquitoes were processed in pools of 6 individuals.

Prior to culturing individuals were washed in a solution of 90% ethanol to remove external adhering bacteria. Individual or pooled mosquitoes were placed in a sterile 1.5 ml tube and crushed with a sterile pellet pestle in a volume of 250 μl of sterile phosphate buffered saline. For plating, a 100 μl aliquot of the homogenate was serially diluted and used for spread plating as diagramed in Fig 2. Mosquito homogenates were plated on LB media with no antibiotics to enumerate the total culturable microbial populations. Mosquito homogenates were also plated on LB plates containing carbenicillin (100 μg/ml), a carboxypenicillin antibiotic with action against peptidoglycan and cell wall synthesis [20]; kanamycin (50 μg/ml), an aminoglycoside antibiotic with action against the bacteria 30S ribosomal subunit [21], and tetracycline (50 μg/ml) a polyketide antibiotic which also targets the bacterial 30S ribosomal subunit [22]. Finally, LB plates were also made with a cocktail of all three antibiotics in combination to identify multiple antibiotic resistant bacteria.

Ten random colonies were isolated and characterized from LB plates to generate a low-resolution characterization of the cultivable microbial populations associated with mosquitoes. For antibiotic resistant colonies, growth characteristics and colony morphology guided selection, in order to recover the broadest diversity of isolates. In this respect, this sampling procedure was designed to gain insights into both the structure of the culturable microbial populations through non-selective culturing, and a broad characterization of the antibiotic resistant bacterial populations within the mosquito microbiome. A schematic diagram of the plating scheme is presented in Fig 2.

## Bacterial isolation, DNA extraction, and PCR of bacterial 16S rRNA genes

All isolates were streaked to single colonies over three generations to obtain pure cultures. Each isolate was then grown in liquid LB media for generating frozen permanent stocks and for DNA isolation. DNA was isolated from a 1 ml aliquot of an overnight culture using the E. Z.N.A. Bacterial DNA Kit (Omega BIO-TEK) with the difficult to lyse bacteria optional protocol. Bacterial 16S rRNA genes were amplified using the universal primers 515F and 806R

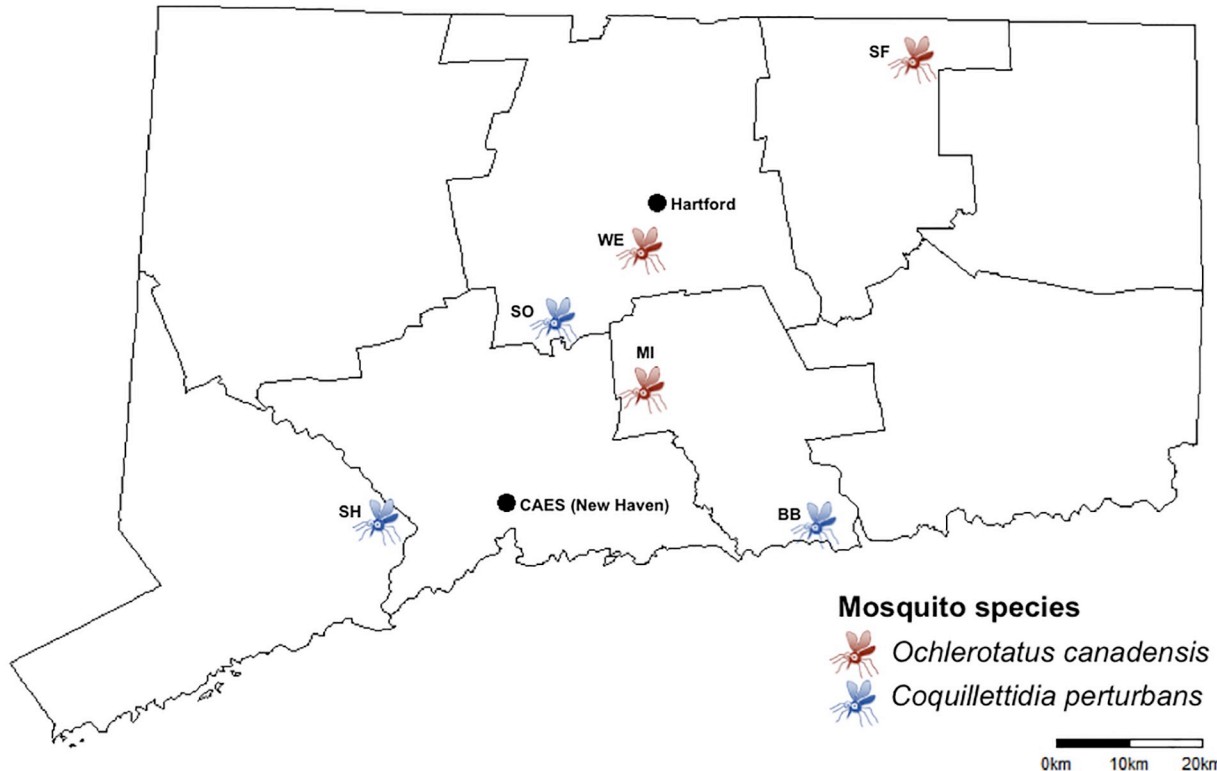

**Fig 1. Mosquito collection sites in the state of Connecticut.** Mosquitoes from two different species were collected from sites around the state. Collection locations are designated by a two-letter code. A full list of sites and their GPS coordinates are provided in S1 Table. The location of the Connecticut Agricultural Experiment Station (CAES) and the state capital (Hartford) are also indicated. A 20 km scale bar is included for a distance reference.

[23,24] and amplification conditions consisting of initial denaturation 95 ˚C, 3 minutes; 30 cycles of the following, 95 ˚C, 45 seconds; 55 ˚C, 45 seconds; 72 ˚C, 1 minute and 45 seconds; followed by a final extension of 10 minutes at 72 ˚C. The resulting amplicons were verified by gel electrophoresis and the PCR reaction was cleaned using the E.Z.N.A. Cycle Pure Kit (Omega BIO-TEK). Purified PCR products were sequenced at the Yale Keck Biotechnology Resource Laboratory by standard Sanger chemistry and the 515F primer as the sequencing primer.

## Fungal isolates

Isolates that were not successfully amplified with the bacterial universal primers were visualized by light microscopy. The majority of the isolates appeared to be fungal cells. For these isolates the fungal LSU gene was amplified with the LR22R [25] and LR3 [26] primer set and identical amplification, cleaning, and sequencing conditions as for PCR described above.

## Bioinformatic analyses

Bacterial 16S rRNA and fungal LSU sequences were quality filtered in the Geneious software package (version 8.1.9) and only sequences with a length of 250 b.p. and quality score greater than 80% were retained. Sequences were assigned to OTUs in the mothur software package (v.1.40.5) [27]. OTUs were assigned with 99% sequence identity. Bacterial 16S rRNA gene sequences were classified with naïve Bayesian Classifier [28] against the SILVA database

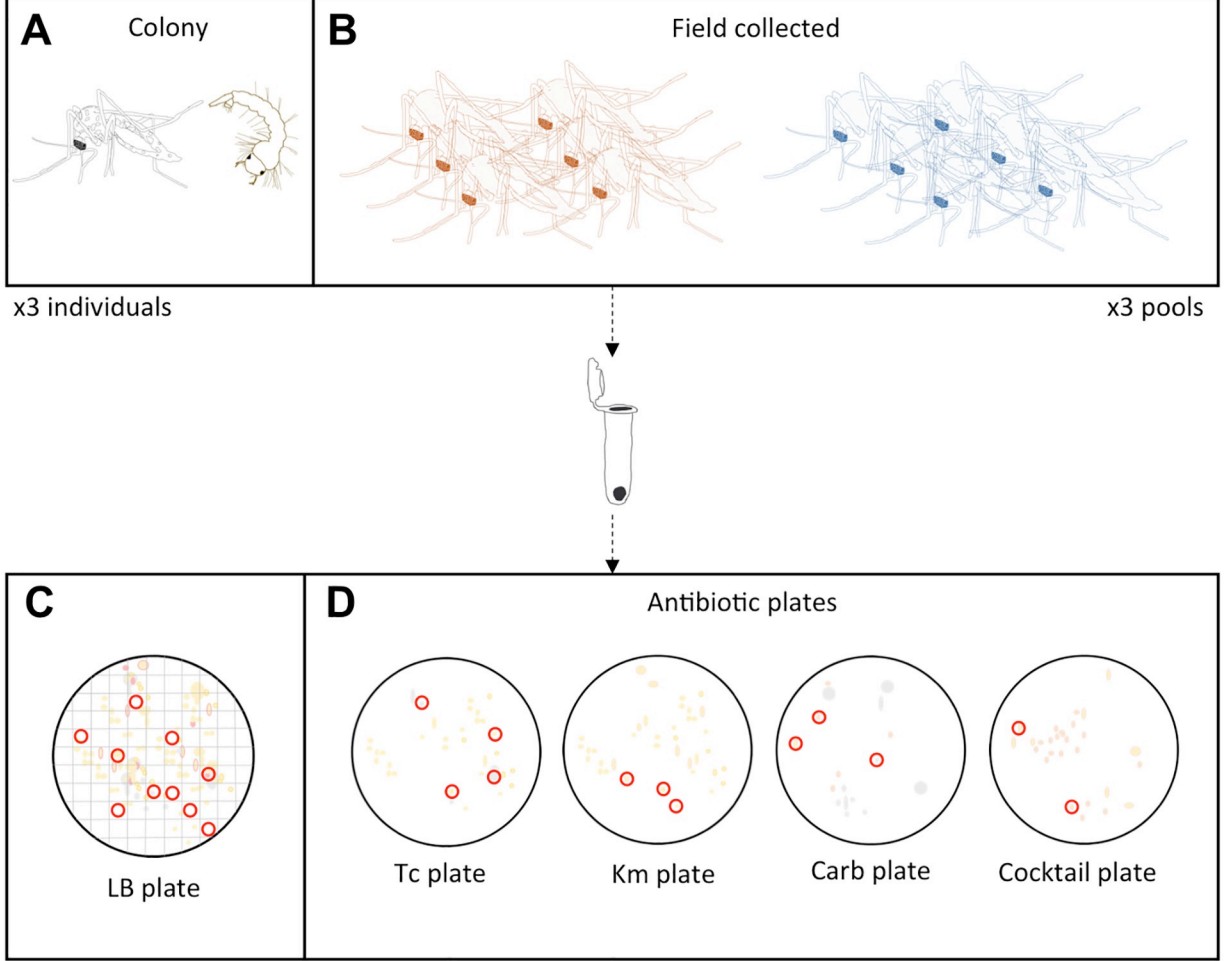

**Fig 2. Schematic diagram of bacterial culturing. A**). Bacteria were isolated from individual colony-reared *Ae. aegypti* larvae and adults. **B**). For field caught mosquitoes bacteria were isolated from a pool of six individual adults. Individuals or pools were homogenized and the homogenates were employed in serial dilutions and plating. **C**). For LB plates a numbered 100 square grid was places under the plate and a random number generator was used to select numbers. If the number drawn and the indicated square contained a colony that colony was selected for isolation, if not another number was drawn. This process was repeated until 10 colonies were selected for isolation. In this manner 10 random colonies were selected to represent a low-resolution profile of the culturable microbial community. **D**). For antibiotic plates (LB media supplemented with antibiotics) colony morphology and growth characteristics guided selection in order to recover the broadest diversity of isolates. Abbreviations: Tc = tetracycline, Km = kanamycin, Carb = carbenicillin.

(version 132; [29]) as implemented in mothur. The fungal LSU sequences were classified against the Fungal LSU Training set [30] also hosted with mothur.

Representative sequences for each OTU (calculated used abundance weighted scoring) were uploaded to the NCBI Nucleotide database with accession numbers MN046275-MN046301 for bacterial 16S rRNA genes and MN046918-MN046927 for fungal 28S rRNA genes.

## Results

### Identification of antibiotic resistance genes in colony reared mosquitoes

The first goal of the project was to test the colony-reared mosquito microbiome for the presence of antibiotic resistance genes by qPCR. A non-overlapping set of seven antibiotic

**Table 1. Antibiotic resistance genes detected in colony larvae and adults.**

| Gene | Antibiotic class | Also detects | Larvae | Adults |
|------|------------------|--------------|--------|--------|
| IMI & NMC-A | Class A beta-lactamase | IMI-2, IMI-3 | **Positive** | Not detected |
| OXA-18 | Class D beta-lactamase | | **Positive** | Not detected |
| OXA-58 Group | Class D beta-lactamase | OXA-96, OXA-97, OXA-164 | **Positive** | Not detected |
| QnrD | Fluorquinolone resistance | | **Positive** | Not detected |
| OXA-60 | Class D beta-lactamase | OXA-60A, OXA-60B, OXA-60c | Not detected | **Positive** |
| SHV(156G) | Class A beta-lactamase | | Not detected | **Positive** |
| aadA1 | Aminoglycoside-resistance | | Not detected | **Positive** |

resistance genes were detected in the colony larvae and adults (Table 1). These data suggest that there was a significant shift in the microbial community between the larvae and adults, as they had no resistance genes in common. The resistance genes detected encompassed four classes of antibiotic resistance, suggesting diverse modes of action in resistance. Thus, these data demonstrate the potential for the colony-reared mosquito microbiome to harbor antibiotic resistance. Based on these data a program to isolate antibiotic resistant bacteria was pursued.

## Plate counts

The viable cell counts were determined for each culture condition. The LB medium with no antibiotics consistently recovered the highest number of CFUs. Antibiotic resistant populations only made up a small fraction of the cells recovered on LB, never exceeding 5% but generally below 1% (Fig 3). In general, carbenicillin and kanamycin resistant populations were most abundant and present in similar counts, with the exception of in the colony-raised larvae, where kanamycin resistant populations were predominant.

## Characterization of bacterial isolates

A total of 198 bacterial isolates were cultured and characterized through 16S rRNA gene sequencing. Clustering of the sequences (99% sequence identity) recovered 27 OTUs (S2 Table). The OTUs represented 4 bacterial phyla (Actinobacteria, Bacteroidetes, Firmicutes, and Proteobacteria). Even with the relatively short sequence fragment (mean 259 b.p.) 20 of the 27 OTUs could be classified to the genus level with confidence scores >80%. Among those OTUs that could be classified to genus, 14 different genera were represented with *Elizabethkingia* and *Morganella* being the most abundant, respectively (S2 Table).

## The cultivable microbiome among mosquitoes

For the bacteria cultured on LB medium, 10 strains were randomly selected for sequencing from each individual mosquito or pool in order to obtain a low-resolution profile of the diversity, abundance, and conservation of bacteria among the mosquitoes (see Fig 2). The patterns of OTU conservation clearly distinguished the colony-raised larvae and adults (Fig 4). The OTUs from the larvae were predominantly from the class Actinobacteria, compared to Gammaproteobacteria for the adults. In fact, no OTUs from the LB plates were shared between the colony larvae and adults, suggesting that the numerically dominant community members differed between the two mosquito developmental stages.

The pattern of OTU conservation also distinguished the colony reared-mosquitoes from the wild caught mosquitoes. The number and taxonomic diversity of OTUs recovered from the wild caught mosquitoes tended to be larger than for the colony mosquitoes. For instance,

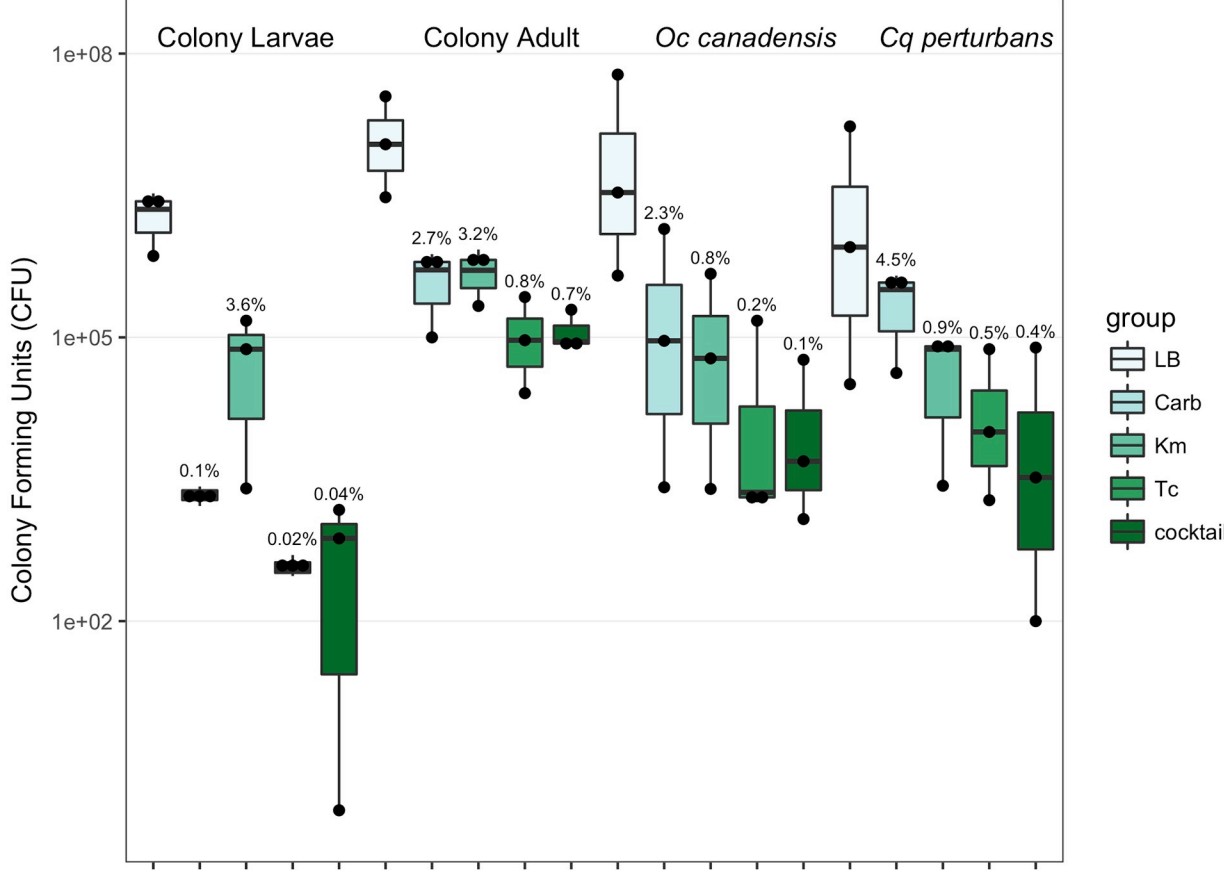

**Fig 3. Bacterial cell counts.** Each bar denotes the recovered CFUs per individual colony-reared mosquito or pools of individuals for the field-caught mosquitoes. Points represent the values for individual replicate plates and the solid bar denotes the mean of the three replicates. Numbers above the boxes indicate the proportion of CFUs recovered on each antibiotic plate as a percentage of the OTUs recovered on the non-selective LB media. The y-axis is log scaled.

bacteria of the classes Bacteroidia and Bacilli were only present in the wild mosquitoes (on the non-selective LB media). The OTUs were also largely unique to either the colony or wild populations. Indeed, only one OTU (Otu06) was found in both the colony and wild caught mosquitoes. Yet, across the different adult mosquitoes the dominant OTUs were generally within the class Gammaproteobacteria, suggesting these bacteria are particularly well adapted for life in association with mosquitoes. Taken together, these observations indicate that even a shallow sampling of the cultivable microbial diversity was sufficient to distinguish the microbial community between colony-raised larvae and adult *Ae. aegypti* mosquitoes, describe differences amongst the colony reared and wild caught mosquitoes, and identify a predominance of Gammaproteobacteria in the mosquito associated bacteria. However, as all culturing was performed on LB media these data likely only represent a proportion of the total bacterial diversity and favor fast growing bacterial species.

## Characterization of antibiotic resistant bacteria

Sampling of antibiotic resistant bacteria targeted diversity rather than abundance in order to recover the broadest selection of antibiotic resistant colonies. The recovered bacteria were identified as belonging to 22 OTUs (Fig 5). Carbenicillin and kanamycin resistant bacteria

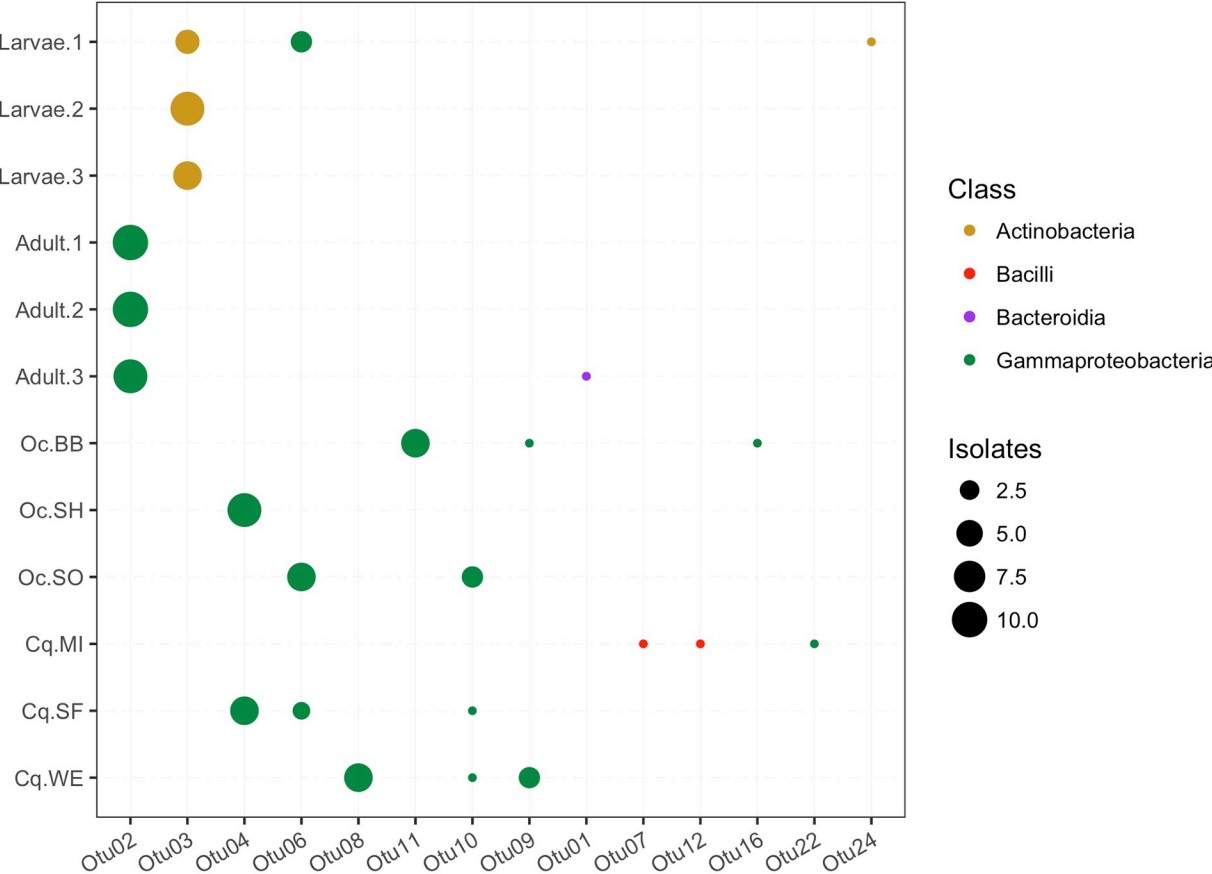

**Fig 4. Composition of the cultivable mosquito microbiome.** Each column represents an OTU detected on LB plates and the rows are the colony-reared individuals or pools of field-caught mosquitoes. The size of each point represents the number of isolates belonging to that OTU from that individual or pool. Each OTU is colored based on the taxonomic class to which each OTU was classified. For the field-caught mosquitoes the labels indicate the species of mosquito (Oc = *Ochlerotatus canadensis*, Cq = *Coquillettidia perturbans*) and the two-letter site code (Fig 1).

were the most commonly recovered, matching observations from the plate counts, indicating carbenicillin and kanamycin resistant isolates were most abundant (Fig 5). Only two OTUs (Otu01 and Otu07) accounted for the bacteria resistant to all three antibiotics (Fig 5).

The number of antibiotic resistant OTUs recovered from the wild populations was greater than for the colony raised *Ae. aegypti* (Fig 5), suggesting a larger diversity of antibiotic resistant bacteria amongst the field caught mosquitoes. Several of the OTUs were also common to multiple pools of mosquitoes caught from different locations suggesting that antibiotic resistant strains were potentially shared between geographically separated mosquito populations. Additionally, there were three OTUs (Otu05, Otu07, Otu09) that were identified in both the colony-reared and field-caught mosquitoes.

## Heterogeneity of antibiotic resistance within an OTU

A potential weakness of this sampling strategy was that isolates from a single antibiotic containing plate might have clustered into an OTU with multi-antibiotic resistant strains. In this respect, the multi-drug resistance of an OTU was not directly tested for every isolate in that OTU. To assay if all of the members of an OTU had identical resistance profiles, the isolates making up Otu01 and Otu07, which contained isolates resistant to all three antibiotics, were

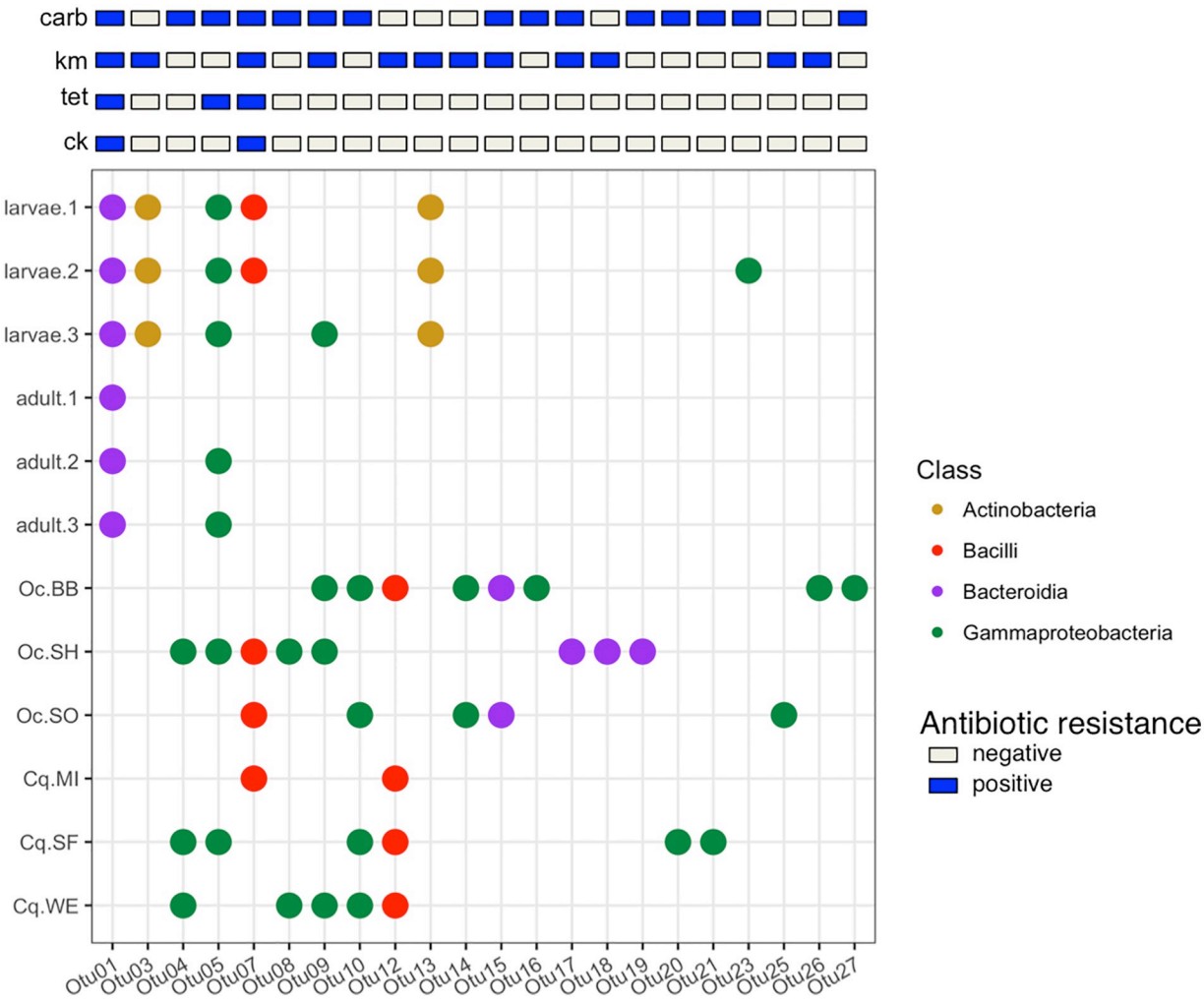

**Fig 5. Composition and conservation of antibiotic resistant OTUs among mosquitoes.** Each column represents an OTU detected on antibiotic plates and the rows are the colony-reared individuals or pools of field-caught mosquitoes. As the sampling was not conducted to assess abundance of isolates the points only reflect presence or absence. The bars above the figure indicate the antibiotic resistance profile of the OTU. The resistance profile was determined such that each OTU was assigned all of the resistances within an OTU. For example, if one isolate in the OTU was from the cocktail plate the resistance profile is indicated as all four resistances although some isolates may have originated from a single antibiotic plate. For the field-caught mosquitoes the labels indicate the species of mosquito (Oc = *Ochlerotatus canadensis*, Cq = *Coquillettidia perturbans*) and the two-letter site code (Fig 1). Abbreviations: Carb = carbenicillin, Tet-tetracycline, Km = kanamycin, Ck = cocktail.

challenged in a multi-resistance assay (Fig 6). For both OTUs a subset of the isolates were resistant to only a single antibiotic, whereas the remainder were resistant to all three (Fig 6). In this respect, OTU membership was not a perfect predictor of an isolate's antibiotic resistance phenotype.

## Fungal isolates

Multiple isolates (30) were determined to be fungi through microscopy, and were subsequently characterized by sequencing of the large subunit (LSU) rRNA gene. The sequences clustered into 10 OTUs (99% sequence identity). Fungal OTUs were classified to both the Basidiomycota and Ascomycota lineages. The OTUs were further classified to five classes with two OTUs only classified to the Domain and Phylum level, respectively (S3 Table). All but two of the isolates

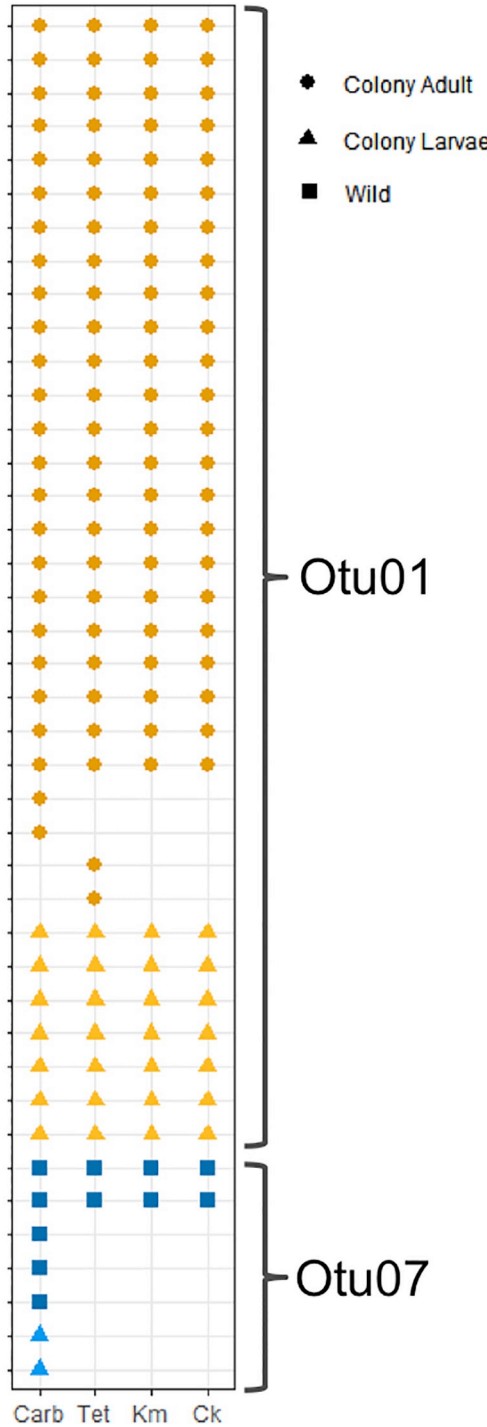

**Fig 6. Heterogeneity of antibiotic resistance in multi-drug resistant OTUs.** Each row represents an isolate that clustered into one of the two multidrug resistant OTUs and the columns represent growth on each antibiotic. The shape of the point represents the source of the isolate. Abbreviations: Carb = carbenicillin, Tet-tetracycline, Km = kanamycin, Ck = cocktail.

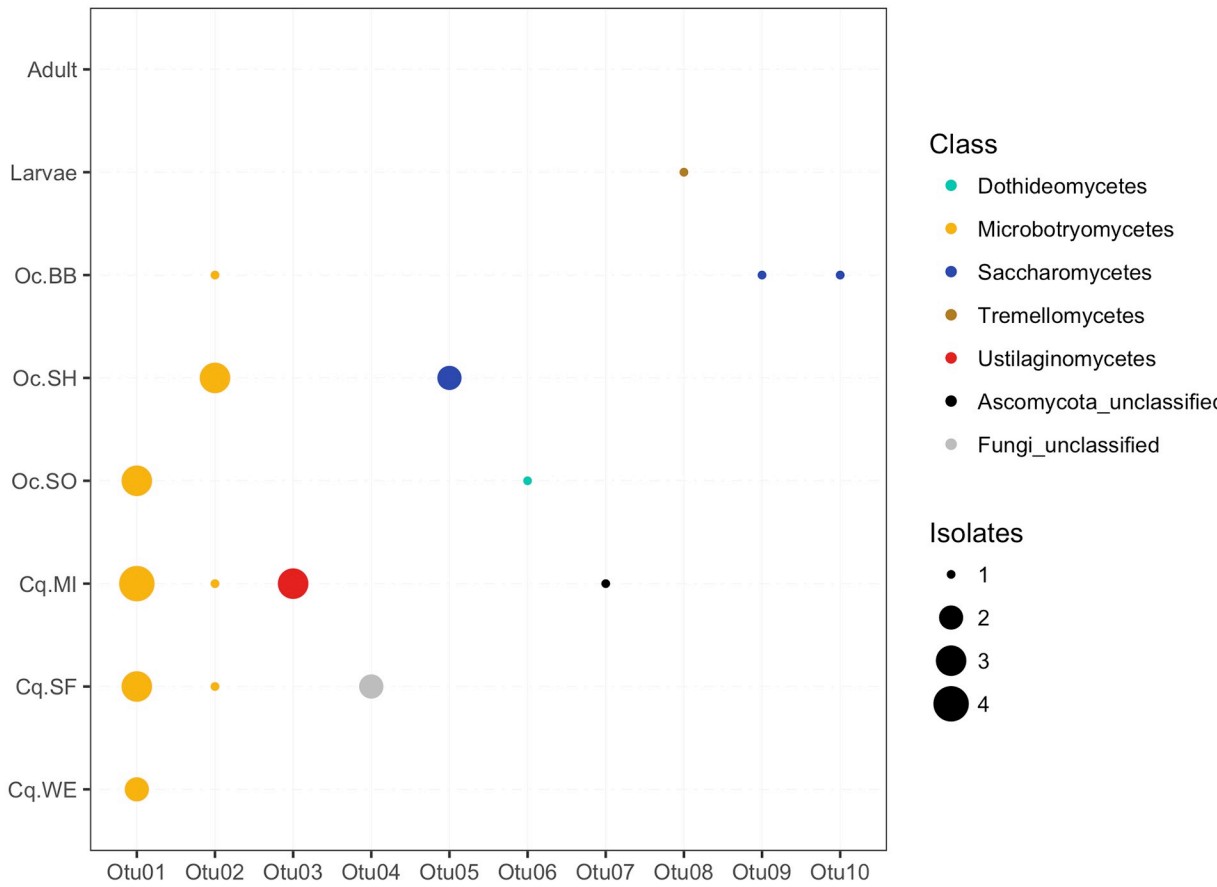

**Fig 7. Composition and conservation of antibiotic resistant OTUs among mosquitoes.** Each column represents an OTU detected on antibiotic plates and the rows are the individuals or pools of field-caught mosquitoes. Because fungi were rare amongst the colony reared mosquitoes all three individuals were combined into a single row. No fungi were isolated from colony adults. The color of the points indicates the fungal class to which the OTU was classified. The size of the points denotes the number of fungal isolates belonging to each OTU.

(both clustering within fungal Otu01) were isolated from antibiotic containing plates. This suggests that the reduction of bacterial numbers by the antibiotics opened up a niche to culture fungal cells. With the exception of the colony-reared adults, fungi were identified in every mosquito pool (Fig 7). Although, only a single fungal isolate was identified in the colony larvae, indicating that fungal abundance was limited in colony-rearing conditions. The wild caught mosquitoes showed an overlap in membership with two fungal OTUs (Otu01 and Otu02) being conserved among the majority of the pools of wild mosquitoes (Fig 7). Both OTUs were classified to the class Microbotryomycetes, suggesting this group of fungi may be common mosquito-associated organisms. The remaining fungal OTUs were all unique to a single mosquito pool, indicating they are less likely to form cosmopolitan stable associations with mosquito hosts.

## Discussion

The majority of research investigating the capacity of insects to harbor antibiotic resistant bacteria has focused on flies and their potential role as vectors to transport antibiotic resistant bacteria from animal production facilities to human populations [31–34]. Additionally, honeybee and various plant feeding insects have also been found to carry antibiotic resistant bacteria

[35,36]. Here we show that antibiotic resistant bacteria, including multiple-antibiotic resistant isolates, are commonly recovered from colony-reared and wild-caught mosquitoes, across three different mosquito genera. These data suggest that the presence of antibiotic resistant bacteria is wide spread among mosquito populations. Yet, the finding of antibiotic resistant bacteria in the colony-reared mosquitoes was surprising, in that the colony has no history of antibiotic treatment. It is increasingly recognized that antibiotic resistant genes can be maintained in the absence of direct selection [37,38]. Thus, previous antibiotic exposure may not always be an indicator for the presence of antibiotic resistance determinants. Mosquitoes are a recognized public health threat as the vector of many diseases. Thus, the finding that mosquitoes also appear to consistently carry antibiotic resistant bacteria could cause some concern. However, as the majority of human pathogens transmitted by mosquitoes are viruses there is a seemingly low risk that mosquitoes will act as a vector of antibiotic resistant bacterial infections. Yet, mosquitoes are mobile, traveling up to a few kilometers per day with longer distance dispersals facilitated by wind and human transport [39], as such they could act as a vector for spreading antibiotic resistant bacteria between environments. Insects have already been identified as a possible route by which antibiotic resistance determinants are spread from rural to urban areas [40].

The most common antibiotic resistant genes detected in the colony-reared mosquito microbiome were beta-lactamases, of which two classes were identified. Beta-lactamases may be among the most common and cosmopolitan resistance genes [41]. Up to 90% of tested soils contained beta-lactamase resistance genes [42], supporting the ubiquitous nature of these genes in the environment. An Aminoglycoside-resistance gene was also detected which could potentially explain the high population numbers resistant to kanamycin (Fig 3). The wide distribution of aminoglycoside resistant genes has significantly reduced the use of these antibiotics in clinical settings [43,44].

Two of the isolates met the requirements to be considered multi-drug resistant, defined as being resistant to at least 3 classes of antibiotics [45]. Thus, those isolates capable of growth on the antibiotic cocktail of the three antibiotics employed in this study meet this criterion. One of the multi antibiotic resistant OTUs (Otu01) was classified to the genus *Elizabethkingia*. Organisms from this genus have been previously been isolated from field-caught mosquitoes from The Gambia, and isolates demonstrate broad spectrum antibiotic resistance, including ampicillin, kanamycin, streptomycin, chloramphenicol, and tetracycline [15]. Here we demonstrate a similar multi-drug resistant phenotype from an *Elizabethkingia*-related isolate from colony-reared *Ae. aegypti*. An observation of note is that isolates that clustered into the same multi-resistant OTU showed heterogeneous antibiotic resistance profiles. For instance, the majority of isolates within Otu01 were resistant to all three antibiotics tested, but 4 (12%) were sensitive to only a single antibiotic (Fig 6). This could be explained by resistance genes being present on plasmids or transposons only shared among subpopulations of an OTU. Similarly, point mutations in resistance genes could only be present in a fraction of the OTU members. Alternatively, the clustering of isolates with differing phenotypes could be due to the short fragment of the 16S rRNA gene sequenced in this study (*c.a.* 259 b.p.). In this regard, longer 16S rRNA gene sequences or multi-locus sequencing may have been better able to distinguish the isolates as different taxa. Yet, these are the same primers used for the Earth Microbiome Project [46] and are likely among the most commonly employed primers for culture-independent microbial diversity assays. So any other study employing these primers would have likely generated similar OTUs. These data highlight that OTU membership can be a flawed predictor for antibiotic resistance phenotype.

Finally, through our efforts to culture antibiotic resistant bacteria we opened up a niche to culture fungal isolates from the mosquitoes. Fungi were more common and abundant among

the wild-caught mosquitoes, suggesting that colony-rearing conditions may not be conducive for mosquito-associated fungi, or at least mosquito associated fungi were not abundant in the laboratory. The finding of conserved fungal OTUs, particularly belonging to the class Micro-botryomycetes (Fig 7) between mosquitoes collected at disparate sites suggests that these fungi form stable relationships with mosquitoes. The majority of research into mosquito-fungal associations has generally focused on entompopathogenic fungi and their role in mosquito control [47–49], whereas the potential role of fungi as commensal flora has been generally neglected. Yet, studies have documented fungal taxa such as *Aspergillus*, *Penicillium*, *Candidia*, and *Fusarium* associated with mosquitoes [4,50,51]. Presumably all of these fungi are not pathogens. One family of fungi, the Harpellales, has been shown to benefit mosquito larval development and thus has been considered to be a commensal [52,53]. But by in large the potential role and beneficial relationships between mosquitoes and fungi has been under studied. Here we show fungi may be conserved microbiome members amongst wild-caught mosquitoes, but their potential role in mosquito biology is essentially unknown.

## Conclusion

The data presented here demonstrate the potential of mosquitoes to act as an environmental reservoir of antibiotic resistant bacteria, including multi-drug resistant strains. Furthermore, the presence of conserved fungal OTUs among field-caught mosquitoes collected from disparate locations points to a role for fungi to associate with mosquitoes. Thus, antibiotic resistant bacteria and fungi should be considered to common members of the mosquito microbiome.

## Supporting information

**S1 Table. Location of the collection sites in the state of Connecticut.** The location of each site is indicated along with the two-letter code that is used to designate the site throughout the manuscript.
(DOCX)

**S2 Table. Taxonomic classification of representative sequences for bacterial OTUs.** *Number of isolates belonging to the OTU. Values in brackets indicate the confidence scores for the classification.
(XLSX)

**S3 Table. Taxonomic classification of representative sequences for fungal OTUs.** *Number of isolates belonging to the OTU. Values in brackets indicate the confidence scores for the classification.
(XLSX)

## Acknowledgments

We would like to thank John Sheppard and the members of the Connecticut Mosquito Trapping and Arbovirus Surveillance Program for collecting and identifying the wild-caught mosquitoes.

## Author Contributions

**Conceptualization:** Doug E. Brackney, Blaire Steven.

**Data curation:** Josephine Hyde, Courtney Gorham.

**Formal analysis:** Josephine Hyde, Courtney Gorham, Blaire Steven.

**Funding acquisition:** Doug E. Brackney, Blaire Steven.

**Investigation:** Courtney Gorham, Blaire Steven.

**Supervision:** Doug E. Brackney.

**Visualization:** Josephine Hyde.

**Writing – original draft:** Josephine Hyde, Doug E. Brackney, Blaire Steven.

**Writing – review & editing:** Josephine Hyde, Doug E. Brackney, Blaire Steven.

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
