## [Decision Letter · Decision Letter 0]

16 Jul 2019

PONE-D-19-16438

Antibiotic resistant bacteria and commensal fungi are common and conserved in the mosquito microbiome

PLOS ONE

Dear Dr Steven,

Thank you for submitting your manuscript to PLOS ONE. After careful consideration, we feel that it has merit but does not fully meet PLOS ONE’s publication criteria as it currently stands. Therefore, we invite you to submit a revised version of the manuscript that addresses the points raised during the review process.

Both reviewers are positive about your manuscript. I leave it p to you if you want to have the fungi included yes or no. Please reply to the other comments.

We would appreciate receiving your revised manuscript by Aug 30 2019 11:59PM. To enhance the reproducibility of your results, we recommend that if applicable you deposit your laboratory protocols in protocols.io, where a protocol can be assigned its own identifier (DOI) such that it can be cited independently in the future. For instructions see: http://journals.plos.org/plosone/s/submission-guidelines#loc-laboratory-protocols

We look forward to receiving your revised manuscript.

Kind regards,

Patrick Butaye, DVM, PhD

Academic Editor

PLOS ONE

Journal Requirements:

1. In your Methods section, please provide additional information regarding the permits you obtained for the work. Please ensure you have included the full name of the authority that approved the field site access and, if no permits were required, a brief statement explaining why.

To comply with PLOS ONE submissions requirements for field studies, please provide the following information in the Methods section of the manuscript and in the “Ethics Statement” field of the submission form (via “Edit Submission”):

a) Provide the name of the authority who issued the permission for each location (for example, the authority responsible for a national park or other protected area of land or sea, the relevant regulatory body concerned with protection of wildlife, etc.). If the study was carried out on private land, please confirm that the owner of the land gave permission to conduct the study on this site.

b) For any locations/activities for which specific permission was not required, please

- i. state clearly that no specific permissions were required for these locations/activities, and provide details on why this is the case

- ii. confirm that the field studies did not involve endangered or protected species

c) For vertebrate studies only, please provide the following additional information:

- i. Full details of collection and sampling methods, including method of sacrifice if applicable

- ii. State whether the vertebrate work was approved by an Institutional Animal Care and Use Committee (IACUC) or equivalent animal ethics committee. If no approval was obtained, please explain why it was not required.

- iii. State clearly whether all sampling procedures and/or experimental manipulations were reviewed or specifically approved as part of obtaining the field permit.

For more information about PLOS ONE submissions requirements for field studies, please refer to http://journals.plos.org/plosone/s/submission-guidelines#loc-animal-research.

Reviewers' comments:

Reviewer's Responses to Questions

**Comments to the Author**

1. Is the manuscript technically sound, and do the data support the conclusions?

Reviewer #1: Yes

Reviewer #2: Yes

2. Has the statistical analysis been performed appropriately and rigorously? 

Reviewer #1: N/A

Reviewer #2: Yes

3. Have the authors made all data underlying the findings in their manuscript fully available?

Reviewer #1: Yes

Reviewer #2: Yes

4. Is the manuscript presented in an intelligible fashion and written in standard English?

Reviewer #1: Yes

Reviewer #2: Yes

5. Review Comments to the Author

Reviewer #1: This reviewer enjoyed very much in reading this excellent manuscript. This work combines the fields of the vector-borne diseases, antimicrobial resistance and microbiology. The manuscript is well written, and the conclusion is well supported by the data. Main comments:

1) I strongly feel like that the portion of commensal fungi can be left out for another independent publication. Please focus on the beauty of this work, antimicrobial resistance and mosquitoes. Do not get distracted by the addition of fungi, in which you did not get any isolates, and cannot characterize the nature of their antimicrobial resistance;

2) For the field-caught mosquitoes, did you try to wash the mosquitoes before performing bacterial isolation etc? There are constant contact between mosquitoes and animal hosts/environmental surfaces etc. The identified microbiome could be on the surface of the mosquitoes, or their inside? Please provide more information, and make meaningful discussion;

3) As indicated in Fig 2 (Schematic diagram of bacterial culturing), you used LB initially, followed by the utilization of different selective media. Please justify this approach. What would you get if you would use the selective media directly without the use of LB media? Even LB is not a type of selective media, the process of LB culture may still provide some preference of certain bacteria which overgrow over others;

4) It is great that the authors used the bioinformatics to identify bacterial OUTs in this work. In the meantime, it would be more clinically more important and meaningful if you could identify the genus level and even species level of the isolated bacterial from the antibiotics-resistant plates. You are very close to provide this valuable information since these isolates are available to you, and you can even ID them by the use of the whole-length 16S rRNA sequencing.

Reviewer #2: Well done paper.

On line 95 I think you are missing the word resistant - it states antibiotic populations and I think you meant to say antibiotic resistant populations. That is the only change that I have.

6. PLOS authors have the option to publish the peer review history of their article (what does this mean?). If published, this will include your full peer review and any attached files.

Reviewer #1: Yes: Chengming Wang

Reviewer #2: Yes: James F Lowe

---

## [Author Response · Author response to Decision Letter 0]

18 Jul 2019

Reviewer #1: 

This reviewer enjoyed very much in reading this excellent manuscript. This work combines the fields of the vector-borne diseases, antimicrobial resistance and microbiology. The manuscript is well written, and the conclusion is well supported by the data. 

Response:

We would like to thank the reviewer for their kind comments. 

Main comments:

1) I strongly feel like that the portion of commensal fungi can be left out for another independent publication. Please focus on the beauty of this work, antimicrobial resistance and mosquitoes. Do not get distracted by the addition of fungi, in which you did not get any isolates, and cannot characterize the nature of their antimicrobial resistance;

Response:

We feel that the fungal data should remain in the current manuscript. Our reasoning is as follows:

These data are based on isolates. As we state in line 356 “through our efforts to culture antibiotic resistant bacteria we opened up a niche to culture fungal isolates from the mosquitoes”. We hypothesize that we recovered these fungal isolated by reducing the bacterial load on the antibiotic plates. We did not actually set out to culture fungi. In this regard our base media (LB) was not optimized for fungal recovery, as such the identification of fungi was a spurious finding in this study, not an objective. This data would not likely be sufficient for an independent manuscript. 

2) For the field-caught mosquitoes, did you try to wash the mosquitoes before performing bacterial isolation etc? There are constant contact between mosquitoes and animal hosts/environmental surfaces etc. The identified microbiome could be on the surface of the mosquitoes, or their inside? Please provide more information, and make meaningful discussion;

Response:

This is something that we took care to control for but failed to mention in the methods. We have revised the methods to read “Prior to culturing individuals were washed in a solution of 90% ethanol to remove external adhering bacteria.” line 148

3) As indicated in Fig 2 (Schematic diagram of bacterial culturing), you used LB initially, followed by the utilization of different selective media. Please justify this approach. What would you get if you would use the selective media directly without the use of LB media? Even LB is not a type of selective media, the process of LB culture may still provide some preference of certain bacteria which overgrow over others;

Response:

The selective plates were LB plates supplemented with antibiotics. The purpose of this was to determine the proportion of antibiotic resistant cells in the “total” population. To avoid confusion we have added “LB media supplemented with antibiotics” to the figure legend of Figure 2 to make it clear that the media formulation is the same between all of the plates. 

We agree with the reviewer that only using LB media likely influenced the bacteria that we recovered. We have added the following “However, as all culturing was performed on LB media these data likely only represent a proportion of the total bacterial diversity and favor fast growing bacterial species” to the results.

4) It is great that the authors used the bioinformatics to identify bacterial OUTs in this work. In the meantime, it would be more clinically more important and meaningful if you could identify the genus level and even species level of the isolated bacterial from the antibiotics-resistant plates. You are very close to provide this valuable information since these isolates are available to you, and you can even ID them by the use of the whole-length 16S rRNA sequencing.

Response:

While we agree with the reviewer that full-length sequencing could improve some of the ability to discern OTU’s, we argue that performing full length sequencing is not warranted for the following reasons.

1. Even full-length 16S rRNA gene sequencing is not generally considered to be sufficient to differentiate species or strains. This level of discrimination is generally performed by multi-locus sequencing, DNA-DNA reassociation kinetics, or whole genome sequencing, which is beyond the scope of this study.

2. Even with the short fragment that we sequenced ~75% of the isolates were classified to genus with >80% confidence (line 232). In this respect, there is likely to be little improvement in our data with full-length gene sequencing.

3. We have dedicated a section of our discussion (line 356) to discuss the potential limitations of using a short sequence fragment, so have already addressed this concern in the manuscript. 

Reviewer #2: Well done paper.

On line 95 I think you are missing the word resistant - it states antibiotic populations and I think you meant to say antibiotic resistant populations. That is the only change that I have.

Response:

The suggested change has been made.

---

## [Editor Report · Decision Letter 1]

1 Aug 2019

Antibiotic resistant bacteria and commensal fungi are common and conserved in the mosquito microbiome

PONE-D-19-16438R1

Dear Dr. Steven,

We are pleased to inform you that your manuscript has been judged scientifically suitable for publication and will be formally accepted for publication once it complies with all outstanding technical requirements.

With kind regards,

Patrick Butaye, DVM, PhD

Academic Editor

PLOS ONE
---

## [Editor Report · Acceptance letter]

5 Aug 2019

PONE-D-19-16438R1 

Antibiotic resistant bacteria and commensal fungi are common and conserved in the mosquito microbiome 

Dear Dr. Steven:

I am pleased to inform you that your manuscript has been deemed suitable for publication in PLOS ONE. Congratulations! Your manuscript is now with our production department. 

With kind regards,

on behalf of

Professor Patrick Butaye 

Academic Editor

PLOS ONE